# Toxicity of Zn-Fe Layered Double Hydroxide to Different Organisms in the Aquatic Environment

**DOI:** 10.3390/molecules26020395

**Published:** 2021-01-13

**Authors:** Olga Koba-Ucun, Tuğba Ölmez Hanci, Idil Arslan-Alaton, Samira Arefi-Oskoui, Alireza Khataee, Mehmet Kobya, Yasin Orooji

**Affiliations:** 1Department of Environmental Engineering, School of Civil Engineering, Istanbul Technical University, 34469 Maslak, Istanbul, Turkey; kobaucun@itu.edu.tr (O.K.-U.); tolmez@itu.edu.tr (T.Ö.H.); 2Research Laboratory of Advanced Water and Wastewater Treatment Processes, Department of Applied Chemistry, Faculty of Chemistry, University of Tabriz, Tabriz 51666-16471, Iran; s.arefi@tabrizu.ac.ir; 3Department of Environmental Engineering, Gebze Technical University, 41400 Kocaeli, Gebze, Turkey; kobya@gtu.edu.tr; 4Department of Environmental Engineering, Kyrgyz-Turkish Manas University, Bishkek 720038, Kyrgyzstan; 5College of Materials Science and Engineering, Nanjing Forestry University, Nanjing 210037, China; yasin@njfu.edu.cn

**Keywords:** layered double hydroxide (LDH) catalysts, *Pseudokirchneriella subcapitata*, *Daphnia magna*, *Spirodela polyrhiza*, *Vibrio fischeri*, surface analysis, bioassays

## Abstract

The application of layered double hydroxide (LDH) nanomaterials as catalysts has attracted great interest due to their unique structural features. It also triggered the need to study their fate and behavior in the aquatic environment. In the present study, Zn-Fe nanolayered double hydroxides (Zn-Fe LDHs) were synthesized using a co-precipitation method and characterized by X-ray diffraction (XRD), Fourier transform infrared spectroscopy (FT-IR), scanning electron microscopy (SEM), and nitrogen adsorption-desorption analyses. The toxicity of the home-made Zn-Fe LDHs catalyst was examined by employing a variety of aquatic organisms from different trophic levels, namely the marine photobacterium *Vibrio fischeri*, the freshwater microalga *Pseudokirchneriella subcapitata*, the freshwater crustacean *Daphnia magna*, and the duckweed *Spirodela polyrhiza*. From the experimental results, it was evident that the acute toxicity of the catalyst depended on the exposure time and type of selected test organism. Zn-Fe LDHs toxicity was also affected by its physical state in suspension, chemical composition, as well as interaction with the bioassay test medium.

## 1. Introduction

Recently, layered double hydroxides (LDHs) which are two-dimensional structured anionic clay materials, have attracted interest in various engineering fields [1]. LDHs are a type of engineered nanomaterial (NM) generally represented by the formula “[M^2+^_1−x_ M^3+^_x_ (OH)_2_]^x+^ (A^n−^) _x/n_·mH_2_O”, where M^2+^ stands for the divalent cation, M^3+^ for the trivalent cation, and A^n−^ for the interlayer anion [2,3,4,5,6]. Their high specific surface area, catalytic ability, anion exchange capability, thermal stability, tunability, as well as flexibility in their interlayer spaces [2,7] render them high performance and unique materials. Hence LDHs found their applications in various fields such as biology [1,6], electrochemistry [6,8,9], photochemistry [10,11], pharmacy and medicine [1,12]. They are also used as additives in polymers, adsorbents in water purification systems [13] and, in particular, as catalytic nanomaterials or catalyst supports [10,14,15,16,17]. Additionally, LDHs reduce catalyst leaching by keeping the pH at alkaline conditions. Owing to their low cost, sustainability, high catalytic activity, power, absorbing nature, and chemical stability LDHs have proven to be suitable materials for environmental remediation applications [18,19]. For instance, Zn-Fe LDHs was reported to have efficient adsorption capacity (equilibrium adsorption = 74.50 mg/g) for diclofenac from aqueous solution [13], while a graphene oxide/Mg-Fe-LDHs composite exhibited high adsorption capacity for heavy metal ions including Pb(II), Cd(II), Cu(II), and Zn(II) [20]. Zn-Fe bifunctional materials were successfully used in the simultaneous removal of pharmaceuticals and arsenic through photochemical oxidation and adsorption from water [21]. Multi-metal Cu-Zn-Fe-LDHs materials were applied in water treatment as heterogeneous Fenton-like oxidation catalysts and adsorbents for acetaminophen and arsenic [22]. It was also reported that Cu-Mg-Fe LDHs exhibited excellent performance for the degradation of ethylbenzene by persulfate activation [23]. In a study by Zhao et al. (2018), Co-Mn LDHs showed excellent performance in organic dye removal via peroxymonosulfate activation [24]. The thermal stability and antibacterial activity against some clinically important bacterial pathogens, namely *Bacillus subtilis*, *Pseudomonas aeruginosa*, *Staphylococcus aureus*, and *Escherichia coli* was proven for a polyacrylonitrile/Zn-Al LDHs nanocomposite [25,26]. Wang et al. (2018) prepared various types of lysozyme modified LDHs which exhibited excellent antibacterial activity and wound healing ability [27].

Despite the above mentioned exciting and promising applications of LDHs, their potential adverse effects require further investigation. There are some basic mechanisms by which NMs such as LDHs may elicit potential toxicity, namely direct interaction with the cell surface, dissolution of toxic elements from nanoparticles, and production of reactive oxygen species (ROS) causing oxidative stress [28,29]. Hence, the selection of appropriate toxicity tests and test organisms are critical issues to obtain reliable and sensitive results in ecotoxicological studies [30,31]. In particular, the use of organisms from different trophic levels is highly recommended to follow-up toxic effects and associated health risks in the form of battery tests [32]. Among the available toxicity tests, the *Vibrio fischeri* (*V. fischeri*) bioluminescence inhibition assay is frequently chosen due to its high reproducibility, speed, and sensitivity towards a wide range of pollutants [33]. On the other hand, green algae are primary producers and key indicators of the environmental balance in aquatic ecosystems. Growth inhibition tests employing unicellular algae are very common and routinely used to determine the toxicity of a variety of pollutants. Among these algal test species, the freshwater microalgae *Pseudokirchneriella subcapitata* (*P. subcapitata*; previously known as *Selenastrum capricornutum*) was found to be the most sensitive one for ecotoxicological studies [34]. Similarly, *Daphnia magna* (*D. magna*) being one of the freshwater crustaceans/invertebrates, have also been defined as routine and standard tools of toxicity test protocols [35,36,37]. They have been used in preliminary screening tests for evaluating the lethal toxicity of different (mainly industrial) chemicals to mammals and humans [38]. Moreover, the toxicity response of higher plants is extremely helpful in monitoring the environmental impact of contaminants [39]. Hence, the *Lemna* bioassay, also known as the duckweed test, is one of the most common, standardized test procedures for ecotoxicological studies involving higher plants [40]. Herein, species from the genus *Lemna*, monocotyledonous free-floating aquatic macrophytes from the *Lemnaceae* family are routinely used [40]. Last but not least, algal, daphnid, and Lemna tests are all Organisation for Economic Co-operation and Development (OECD) assays as well as legislator tests, for example, Registration, Evaluation, Authorization, and Restriction of Chemicals (REACH) [40]. For all of these reasons, the above-mentioned test organisms were employed in the present study.

Although a variety of physicochemical properties of novel NMs have already been investigated by different authors, their ecotoxicological impacts have not been examined in detail so far. Considering the above-mentioned gap in the scientific literature, the present work aimed to examine the toxicity of a home-made Ze-Fe LDHs nanocatalyst by employing aquatic organisms from different trophic levels. For this purpose, firstly, Zn-Fe LDHs catalyst was synthesized with a facile and cost-effective co-precipitation method and characterized by SEM, XRD, FT-IR, and nitrogen adsorption-desorption analyses. In the second part of the study, possible environmental risks of the Zn-Fe LDHs catalyst were tested using the marine photobacterium *V. fischeri*, the freshwater microalga *P. subcapitata*, the freshwater crustacean *D. magna* and the duckweed *S. polyrhiza* forming a battery assay. To the best of our knowledge, this is the first attempt to evaluate the multi-trophic acute toxicity of the Zn-Fe LDHs catalyst.

## 2. Results and Discussion

### 2.1. Zn-Fe LDHs Characterization

The XRD pattern of the synthesized Zn-Fe LDHs is represented in Figure 1a. The peaks centered at 2θ of 12.1°, 23.3°, 34.5°, 39.0°, 45.9°, 59.7°, and 61.2° can be attributed to the 003, 006, 009, 015, 018, 110, 113, and 113 crystalline planes of the synthesized sample, indicating the successful synthesis of LDHs [15]. In addition, the average crystallite size of the synthesized LDHs was determined as approximately 15 nm using Debye-Sherer’s equation [41].

The FT-IR spectrum of the synthesized Zn-Fe LDHs was recorded in the wavenumber range of 500–4000 cm^−1^ to investigate the functional groups (See Figure 1b). The absorption peaks detected in the wavenumber lower than 1000 cm^−1^ can be ascribed to the vibration of the O-M-O and M-O groups (M: Zn^2+^ and Fe^3+^ cations) [42]. The absorption peaks centered at 1419 cm^−1^ and 1475 cm^−1^ are assigned to the carbonate (CO_3_^2−^) anions placed in the interlayer region of LDHs [43]. The source of the carbonate anions in the interlayer region of LDHs can be attributed to the dissolution of CO_2_ molecules in distilled water used for synthesis [15]. The peak centered at about 1600 cm^−1^ is due to the bending vibration of water molecules that exist in the interlayer region of LDHs [44]. The strong and broad band observed in the wavenumber of 3500 cm^−1^ can be assigned to the O-H groups attached to the layer surface of the LDHs and the O-H groups of the interlayered water molecules [45].

The SEM images of the synthesized Zn-Fe LDHs are represented in Figure 2. Layered morphology with an average thickness of 45 ± 0.6 nm can be seen in the SEM images.

The nitrogen adsorption-desorption isotherm of the prepared Zn-Fe LDHs is shown in Figure 3. According to the IUPAC classification, the synthesized LDHs represented type IV isotherm, confirming the mesoporous structure of the synthesized sample. Based on the Brunauer-Emmett-Teller (BET) analysis, the specific surface area of the synthesized LDHs was determined 46.9 m^2^/g.

### 2.2. Acute Toxicity Assessment

The toxicity of NMs stems from the promotion of ROS production inducing cellular oxidative stress, or from direct interactions between these materials, their dissolution products (released metal ions), and proteins. Oxidative stress may cause the damage of lipids, membrane, carbohydrates, and deoxyribonucleic acid (DNA) via formation of disulfide, oxidation of methionine, and carbonylation of amino acid residues promoting the accumulation of nanoparticles [29,46]. The surface coating can cause inhibition of photosynthesis in algal tests; besides, nutrient uptake or movement and is also considered to be one of the possible toxicity routes of NMs [46,47]. The aquatic toxicity of the Zn-Fe LDHs towards test organisms being selected from different trophic levels is presented and discussed in the following sections.

#### 2.2.1. *V. fischeri*

As aforementioned, the *V. fischeri* luminescence inhibition assay is one of the most frequently used ecotoxicological tests for the in vitro toxicity evaluation of different contaminants [48]. The bioluminescence of the Gram-negative marine bacterium *V. fischeri* is a result of biochemical reactions where a reduced flavin mononucleotide, a long-chain fatty acid aldehyde, and luciferase are involved. This process uses nicotinamide adenine dinucleotide hydrogen (NADH) as a cofactor for the reactions and is related to the central metabolism of the bacterium. Consequently, the inhibition of V. fischeri bioluminescence strongly correlates with the toxicity of the test sample [49,50].

Figure 4 displays the toxicity of Zn-Fe LDHs towards *V. fischeri*. Three catalyst preparation times, namely t = 24, 3, and 1 h were evaluated with *V. fischeri* to question whether the toxic response depends on the sample preparation time. The appearance of all Zn-Fe LDHs samples changed after stirring for 24 h and a thick white film formed on their surface. Moreover, all samples (Zn-Fe LDHs concentrations = 0.10–2.00 g/L) showed an increase in pH to ≈ 8.4 and resistance to pH adjustment making pH re-adjustment to the working pH (7.0 ± 0.2) rather difficult. As a result, only the catalyst concentration of 0.10 g/L could be tested for 24 h-prepared Zn-Fe LDHs samples. Under these test conditions, 24.4 ± 1.2% and 68.5 ± 0.8% relative inhibitions were observed with respect to the toxicant-free control for an exposure time of t = 15 and 30 min, respectively.

As has been reported previously, NMs can be agglomerated into microscopic particles in a high ionic strength medium as in the present case, inducing a change in their morphology and instability in surface properties [51]. Therefore, modification of the NMs’ structural properties after dispersion into the test media should be considered when evaluating toxicity. Based on the results obtained for 24 h-prepared toxicity samples, the experimental procedure was modified and the sample preparation time was reduced to t = 3 and 1 h. As can be seen from Figure 4a, after 15 min of exposure, only the concentrated samples (= 0.50, 1.00, and 2.00 g/L) showed an inhibitory effect on luminescence intensity. The lowest observed effect (a relative inhibition of 15.8 ± 1.7%) was found for 0.50 g/L catalyst. On the other hand, for the exposure time of 30 min, appreciable inhibition was observed for all tested concentrations. The EC_50_ values for the 3 h-prepared Zn-Fe LDHs samples were calculated as 1.20 ± 0.12 g/L and 0.32 ± 0.03 g/L for incubation times of 15 and 30 min, respectively. As can be seen in Figure 4b, the highest luminescence inhibition was observed for the Zn-Fe LDHs sample being prepared in 1 h at the concentration range of 0.05–0.20 g/L. The percent relative inhibition of 0.20 g/L catalyst was found as 28.9 ± 0.1% (instead of 0% for 3 h-prepared sample) and 66.0 ± 0.1% (instead of 33.8 ± 0.6% for 3 h-prepared sample) for an exposure time of 15 and 30 min, respectively. The EC_50_ value was calculated as 0.31 ± 0.03 g/L for t = 15 min, whereas an EC_50_ value of 0.11 ± 0.01 g/L was obtained for t = 30 min. Sample preparation time affected the toxicity response due to effects such as metal ion release and changes in catalyst structure after the contact with the test medium.

In previous work, it was demonstrated that nano-ZnO toxicity (EC_50_ = 1.17 mg/L) was possibly caused by Zn^2+^ release [52]. Fe^3+^ also proved to be very toxic towards *V. fischeri* (EC_50_ = 0.44 mg /L) [53]. A similar trend was also observed for another gram-negative bacteria, recombinant luminescent *E. coli*, where 30-min EC_50_ values were found as 8.5 mg Fe^2+^/L and 1.3 mg Fe^3+^/L [54] speaking for a toxicity effect caused by metal ions and possible metal ion leaching from Zn-Fe LDHs catalyst into the test medium.

#### 2.2.2. *P. subcapitata*

Several interactions between microalgae and NMs can induce toxic response including shading/reduction of captured light, metal ions being released from NMs, oxidative stress, adsorption, absorption, and disruption of microalgae barriers [55]. The algal bioassay principle is based on growth rate inhibition being caused by different contaminants.

In the present study, Zn-Fe LDHs catalyst was found to be toxic towards *P. subcapitata* even at the lowest test concentration of 0.10 g/L. These experimental results are in agreement with previously published data for Zn-containing nanocatalysts. Aruoja et al. (2009) reported that bulk and nano-ZnO were quite toxic towards *P. subcapitata* with EC_50_ values of 0.037 and 0.047 mg/L for the bulk and nano-ZnO, respectively [56]. Another study conducted by Franklin et al. (2007), where the effect of ZnCl_2_ on *P. subcapitata* was examined, reported an EC_50_ value of 0.060 mg/L [57]. Exposure of *Scenedesmus quadricauda*, typical freshwater green algae, to Cu-Mg-Fe LDHs resulted in serious growth inhibition at an incubation time of 72 h (EC_50_ = 10 mg/L) in another work [58]. It should be mentioned here that aggregation and morphological changes of the Zn-Fe LDHs catalyst were observed during the bioassay which could explain the toxic response. Thus, to gain further insight into the toxicity pathway and potential impacts of Zn-Fe LDH catalyst on microalgae, characterization of the LDHs before and after exposure to the test medium could be important.

#### 2.2.3. *D. magna*

Among the invertebrates, *D. magna* as a representative of zooplankton and a key group for the food web structure has often been used in toxicity tests [47]. In addition to ROS-mediated toxic effects of NMs, “biological surface coating” (i.e., the attachment or adsorption of NMs on the outer surface of the organisms) is suggested as a potential toxicity trigger for test organisms like *D. magna* [46]. Figure 5 depicts the effect of Zn-Fe LDHs catalyst concentration (0.05–0.20 g/L) on *D. magna* response for t = 24 and 48 h. As obvious from Figure 5, 10 ± 2% and 65 ± 20% mortalities were observed for 0.05 and 0.20 g/L catalyst t = 24 h, respectively. Mortalities were higher compared to the test control for all tested concentrations at t = 48 h; e.g., 35 ± 20% for 0.05 g/L and 75 ± 10% for 0.20 g/L. Similar to *V. fischeri*, *D. magna* toxicity of Zn-Fe LDHs also depended on the exposure time; toxicity of the Zn-Fe LDHs catalyst increased with prolonged test durations. It was observed that Zn-Fe LDHs catalyst particles adhered to the antennae of the daphnids when higher catalyst concentrations and longer incubation times were applied. Despite external surface adhesion, no catalyst was seen in the gut tract of the test organisms. The EC_50_ concentrations of Zn-Fe LDHs to *D. magna* were calculated as 0.17 ± 0.02 and 0.10 ± 0.01 g/L for 24 and 48 h of exposure, respectively.

Studies devoted to the toxicity of ZnO-NMs towards invertebrates and vertebrates reported that zinc ions released from ZnO-NMs rather than the particle size are the main factor of toxicity. In the study of Bacchetta et al. (2016), the 48 h LC_50_ was 1.02 mg/L and 1.10 mg/L for 30 nm and 80–100 nm ZnO catalyst, respectively [59]. Blinova et al. (2010) reported that the toxicity of CuO NMs in river water is related to the release of dissolved metal ions [60]. On the other hand, the acute toxicity of TiO_2_ and Fe_3_O_4_ in daphnids, for instance, was attributed to a physical inhibition of molting, ultimately inducing death [46,47,61].

#### 2.2.4. *S. polyrhiza*

As mentioned above, the duckweed test is one of the most recognized and routine higher plant toxicity tests for assessing the impacts of contaminants on the aquatic environment. Species from monocotyledonous free-floating aquatic macrophytes of the Lemnaceae family are frequently being used in ecotoxicology studies. The simple anatomy and ease with which they can be handled make them ideal test organisms. However, due to the variety of exposure routes, these higher plants demonstrate a range of sensitivities to the toxic pollutants [40]. In this work, a 72 h-growth inhibition test [62] was conducted for the Zn-Fe LDHs catalyst to elucidate its inhibitory effect towards *S. polyrhiza* at a concentration range of 0.075 to 2.00 g/L (See Appendix A and Figure 6). As evident from Figure 6, a fair inhibition (32 ± 6%) was evidenced for the lowest studied Zn-Fe LDHs concentration (0.075 g/L). Increasing the catalyst concentration to 0.50 and 2.00 g/L led to a gradual increase in *S. polyrhiza* growth inhibition from 32 ± 6% to 41 ± 14% and 66 ± 12%, respectively. Based on these results, EC_50_ values were calculated at 0.81 ± 0.19 g/L for an exposure time of 72 h. At the end of the test duration, a pale-yellow color of the fronds was visually observed for all investigated Zn-Fe LDHs catalyst concentrations (Appendix A), which was considered to be indicative of metal dissolution from the catalyst causing a morphological change of the catalyst. Moreover, the formation of the additional fronds (more than two) was not observed at the highest tested catalyst concentration of 2.00 g/L (Appendix A).

All acute toxicity test results obtained with different test organisms were summarized in Appendix A. From Appendix A it is evident that the catalyst sample is highly toxic (inhibitory) towards the selected test organisms in its typical working (application range) range and the toxic response increases with extended incubation time which would be an important remark to applicants of nanomaterials.

## 3. Materials and Methods

### 3.1. Synthesis and Characterization

Iron (III) chloride hexahydrate (FeCl_3_.6H_2_O), sodium hydroxide (NaOH), and zinc chloride (ZnCl_2_) were purchased from Merck Co., Gernsheim, Germany. The facile co-precipitation method was used to synthesize the Zn-Fe LDHs. In the first step, 1.00 mmol of FeCl_3_.6H_2_O and 3.00 mmol of ZnCl_3_ were dissolved in distilled water. Then, the pH of the solution was adjusted to pH 8 by dropwise addition of NaOH solution under a nitrogen atmosphere and vigorous stirring. The resultant mixture was further stirred for 24 h under a nitrogen atmosphere at room temperature. Finally, the synthesized LDHs particles were collected and separated by centrifugation and dried at 50 °C for 5 h after washing with distilled water.

A Philips X-ray diffractometer with a model of PW1730 (Amsterdam, the Netherlands) was used to study the crystalline structure of the synthesized Zn-Fe LDHs. The morphology of the synthesized sample was investigated using a Tescan scanning electron microscope with a model of MIRA3 (Brno, Czech Republic). The thickness of the synthesized Zn-Fe LDHs was determined using SEM images and Digimizer software, and the average thickness was reported. A Bruker Fourier transform infrared (FT-IR) spectrometer with a model of Tensor (Hamburg, Germany) was used to determine the functional groups of the synthesized LDHs. The nitrogen adsorption-desorption isotherm of the synthesized Zn-Fe LDHs was determined using the BELSORP device with a model of Mini II (BEL, Osaka, Japan).

### 3.2. Acute Toxicity Assessment

#### 3.2.1. Preparation of Toxicity Samples

The samples for the toxicity tests were prepared by the addition of an appropriate amount of the Zn-Fe LDHs into the culturing medium for each test organism. The wide concentration ranges of Zn-Fe LDHs used in this work were chosen based on related applications reported in the scientific literature [10,12,13,14,15]. Appropriate volumes specified for the toxicity tests were prepared by adding the Zn-Fe LDHs catalyst into the test medium through an automatic pipette forming catalyst + test medium suspensions. All test samples were continuously stirred with a magnetic stirrer before and during their transfer into the test vials to ensure that the same concentration of catalyst particles was taken for each replicate. The suspensions remained stable during the tests and no ultrasonication was needed for sample mixing and preparation. Toxicity tests were conducted using commercially available test kits. Algaltoxkit F, Daphtoxkit F, and Duckweed Toxkit F were purchased from MicroBioTests Inc. company (Gent, Belgium and Masku, Finland), whereas the test kits of the marine photobacteria were obtained from Aboatox-company (BioTox^TM^, Aboatox Oy, Masku, Finland). Appendix A gives information about all toxicity test procedures used in this study.

All data were presented as means ± standard deviations. The standard deviation for each toxicity test is reported in the form of error bars in the figures. Duplicates, triplicates, pentaplicates, and eight replicates were used to carry out the bioassays with *V. fischeri*, *P. subcapitata*, *D. magna*, and *S. polyrhiza*, respectively, depending on the test procedure. Statistical significance and quality criteria were tested using the software provided by Aboatox (Gent, Belgium and Masku, Finland) and MicroBioTests Inc. companies. The EC_50_ (= median effective concentration) defined herein as the effective Zn-Fe LDHs catalyst concentration causing 50% reduction in the bioluminescence, growth and/or death/immobilization were calculated using specific guidelines for each test organisms. The significance level in all calculations was set at *p* < 0.05.

#### 3.2.2. *V. fischeri*

The acute toxicity as percent relative bioluminescence inhibition towards the photobacterium *V. fischeri* was measured with a commercial bioassay kit according to the ISO 11348-3 test protocol [63]. Briefly, catalysts were dissolved in 2% *w*/*v* NaCl solution and all the samples’ pH values were adjusted to a fixed value of 7.0 ± 0.2. The lyophilized bacteria with reconstituted reagents were equilibrated at +4 °C for at least 30 min and then stabilized at +15 °C for at least 30 min before pipetting the bacteria suspension into the samples. For each dilution level two parallel samples were prepared and the luminescence intensity was recorded in all test tubes, including controls, after 15 min and 30 min. The test procedure was as follows; firstly, all samples (concentration range = 0.10–2.00 g/L) were prepared according to the assay procedure and left for stirring overnight for 24 h. However, since the catalyst was in particle form and might aggregate after a while, the sample preparation time was reduced from 24 to 3 and 1 h and a lower catalyst concentration range of 0.05–0.20 g/L was selected. In order to be sure that no positive intensity inhibition was observed due to precipitation/aggregation of the catalyst particles, all samples were mixed with the pipette tip just before each luminescence measurement.

#### 3.2.3. *P. subcapitata*

The algal growth inhibition bioassay was performed using freshwater microalgae stock culture *P. subcapitata* from the test kit. The toxicity test was based on percent relative growth inhibition towards *P. subcapitata* and performed according to the ISO 8692 test protocol [64]. Test conditions were set as T= 24 ± 1 °C and pH= 8.1 ± 0.2, and test organisms were exposed to continuous side illumination with cool white light (5000–6000 lux). Experiments were performed in triplicates in 25 mL incubation vials with an initial algal cell count of around 10,000 cells/mL. Algal biomass (optical density) measurements were carried out at t = 24, 48 and 72 h for catalyst concentrations in the range of 0.10–2.00 g/L.

#### 3.2.4. *D. magna*

The dormant eggs (ephippia) of the freshwater crustacean *D. magna* were used in the toxicity bioassay. Acute toxicity tests were based on percent death and/or immobilization rates being observed for *D. magna* after an incubation time of t = 24 and 48 h according to the Standard Operational Procedure of Daphtoxkit F based on the OECD 202 guideline [37,65]. According to the standard test protocol, before each test, ephippias were hatched within 72 h and thereafter the neonates (newborn daphnia) were fed with the unicellular alga *Spirulina* sp. (also provided with the test kit) for 2 h. Briefly, five daphnid neonates per test cell were incubated in the dark at T = 20 ± 2 °C for up to t = 48 h in four replicates. Daphnids were exposed to a catalyst concentration range of 0.05–2.00 g/L. The quality criteria of the tests were completely fulfilled as in the test control immobilization of daphnids did not exceed 10%. The number of immobilized daphnids (the toxicity endpoint) was determined at t = 24 and 48 h.

#### 3.2.5. *S. polyrhiza*

The *S. polyrhiza* toxicity bioassay test is based on measuring the decrease (or absence) of growth of the germinated “turions” (dormant vegetative stages, vegetative buds), after 72 h of exposure in comparison to the test control. The experimental procedure was based on ISO 20227 test protocol [62]. Briefly, the vegetative buds were firstly incubated for three days at 25 °C in a petri dish under continuous illumination with 6000 lux to obtain germinated turions. The pH of the growth medium was adjusted to 5.5 ± 0.2. The germinated turions were transferred into each of the 48 cups of the multi-well test plate containing controls and different concentrations of the catalysts. “Digital pictures” of the test plates were taken to measure the size (area) of the small “first frond” of the duckweeds in the test cups at the beginning and after t = 72 h of exposure. The test areas were measured with the “Image J” software [66]. Eight replicates were used for toxicity evaluation. Validity criteria of the tests were completely fulfilled as the mean growth of the first fronds in the cups of the control row after t = 72 h incubation at 25 °C and under 6000 lux illumination were at least 10 mm^2^. If more than one frond was developed in the same cup, only the largest of these was considered.

## 4. Conclusions

In the present study, Zn-Fe nanolayered double hydroxide (Zn-Fe LDHs) was synthesized, characterized, and comparatively examined for its toxic impacts with four different bioassays using test organisms from the different trophic levels. Firstly, the Zn-Fe LDHs catalyst was prepared using a co-precipitation method. The XRD pattern of the catalyst material confirmed the successful synthesis of Zn-Fe LDHs with an average crystallite size of 15 nm. Layered morphology with an average thickness of 45 nm was observed in the SEM images. The synthesized Zn-Fe LDHs revealed a type IV nitrogen adsorption-desorption isotherm, indicating the mesoporous structure. Besides, the specific surface area of the Zn-Fe LDHs was obtained as 46.9 m^2^/g using BET analysis. A comprehensive acute toxicity assessment was also undertaken for the home-made Zn-Fe LDHs catalyst. Results demonstrated a concentration and preparation time-dependent response for all tested organisms. *P. subcapitata* (complete inhibition at all tested catalyst concentrations) was found to be most sensitive to the Zn-Fe LDHs catalyst, whereas *S. polyrhiza* (EC_50_ = 0.81 ± 0.19 g/L) appeared to be the least sensitive test organism. *V. fischeri* toxicity results highlighted the importance of “preparation time” of the Zn-Fe LDHs catalyst samples before toxicity analysis. The EC_50_ values calculated for the 1 h-prepared catalyst test solutions (EC_50,15 min_ = 0.31 ± 0.03 g/L and EC_50,30 min_ = 0.11 ± 0.01 g/L) were lower than that of the 3 h-prepared solutions (EC_50,15 min_ = 1.20 ± 0.12 g/L and EC_50,30_ min = 0.32 ± 0.03 g/L). EC_50_ values for *D. magna* were detected as 0.17 ± 0.02 g/L and 0.10 ± 0.01 g/L for an incubation time of 24 and 48 h, respectively. The present study highlighted the importance of combining biotests with chemical analysis to provide a more complete picture of the ecotoxicological effects of engineered nanomaterials as well as structure-toxicity relationships. From the study, it could be concluded that several factors affected aquatic toxicity findings; the exposure time together with the type of test organism and physicochemical properties of the standard bioassay test medium all have to be considered when assessing the toxicity of engineered nanomaterials since these might affect the dispersibility, aggregation, sedimentation, and the dissolution of the tested catalyst samples.

## Figures and Tables

**Figure 1 molecules-26-00395-f001:**
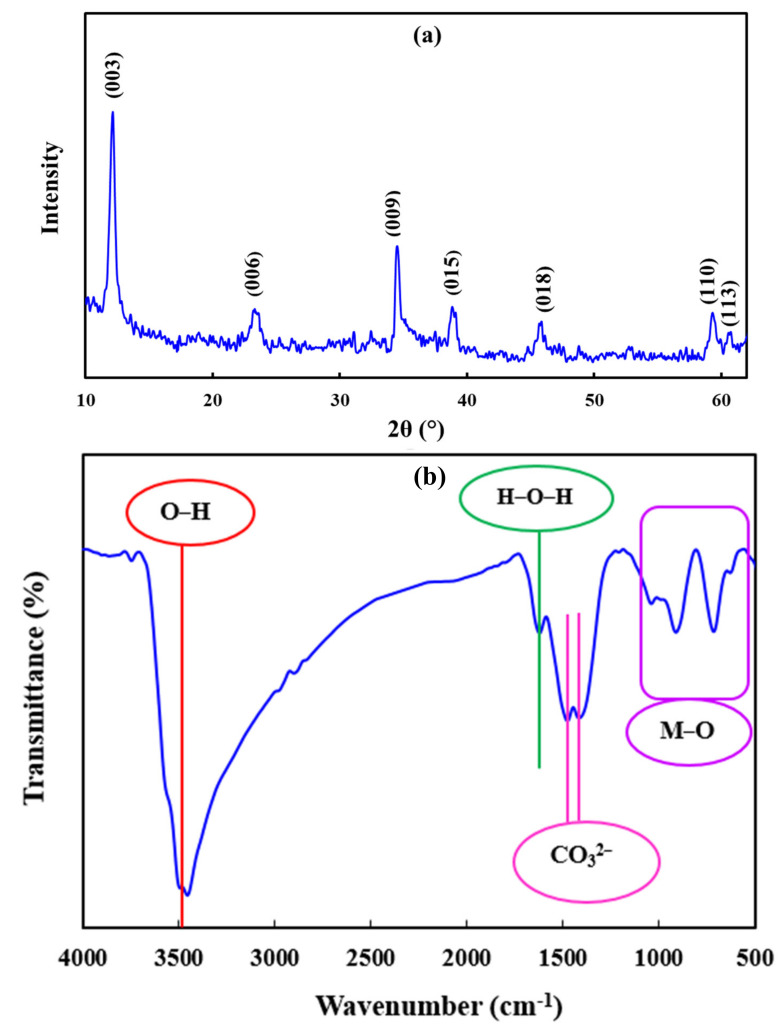
XRD pattern of nanolayered Zn-Fe LDHs (**a**); FT-IR spectrum of nanolayered Zn-Fe LDHs (**b**).

**Figure 2 molecules-26-00395-f002:**
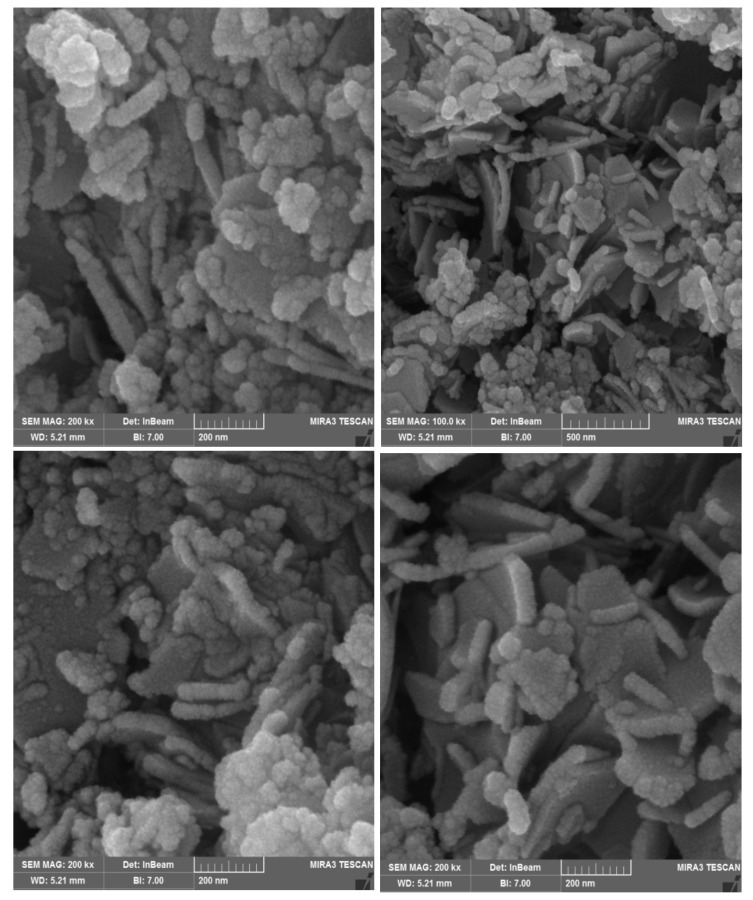
SEM images of the Zn-Fe LDHs.

**Figure 3 molecules-26-00395-f003:**
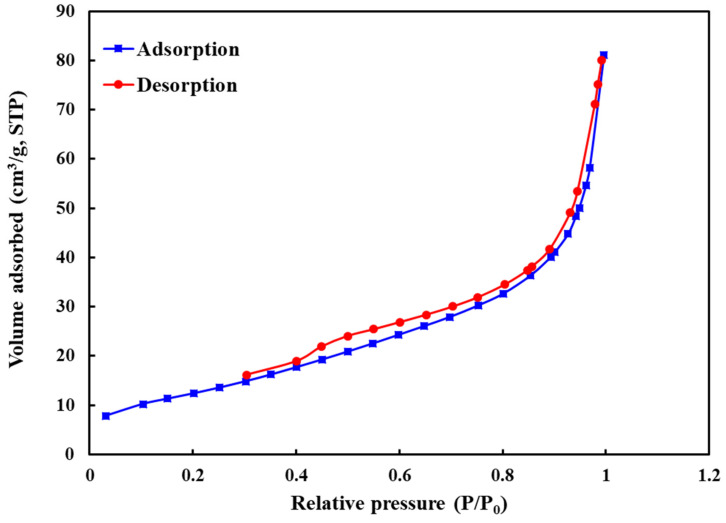
Nitrogen adsorption-desorption isotherm of Zn-Fe LDHs.

**Figure 4 molecules-26-00395-f004:**
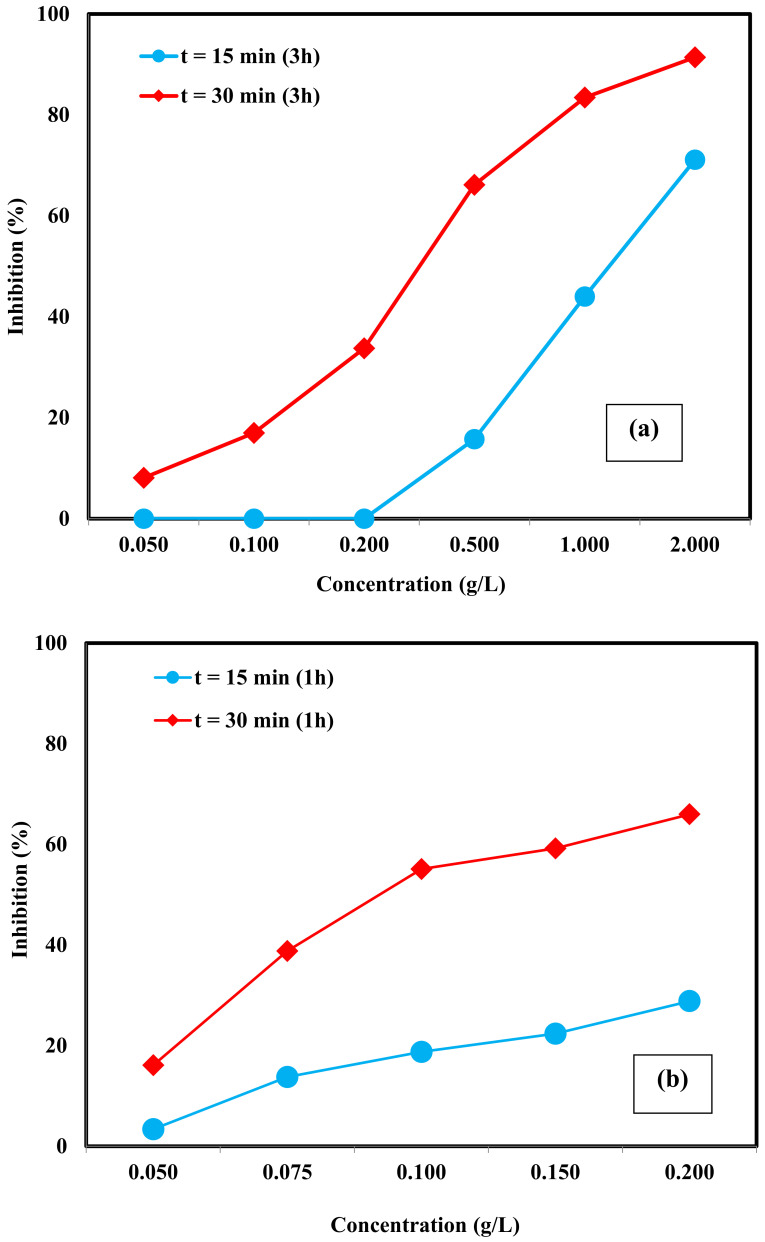
Mean relative inhibition (± SD; *n* = 3) towards *V. fischeri* after 15 and 30 min of exposure to Zn-Fe LDHs toxicity test samples prepared within 3 h (**a**), and 1 h (**b**).

**Figure 5 molecules-26-00395-f005:**
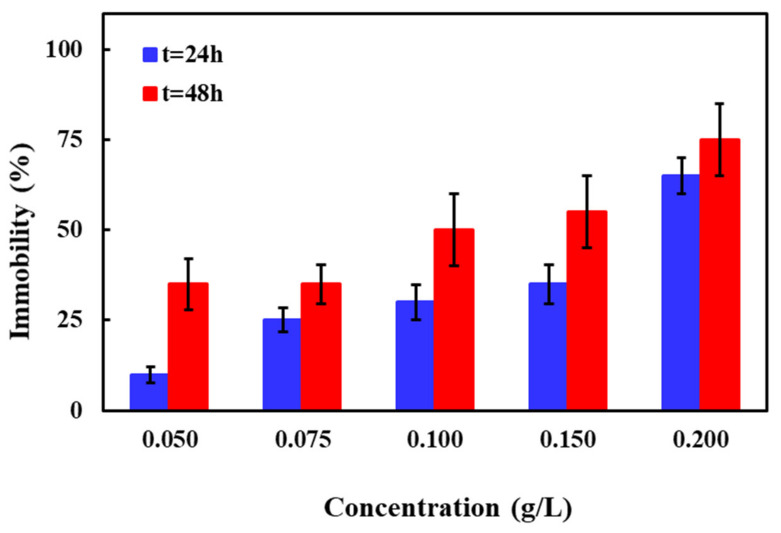
Mean death/immobilization (± SD; *n* = 5) of *D. magna* after 24 and 48 h of exposure to concentrations of 0.05, 0.075, 0.10, 0.15, and 0.20 g/L Zn-Fe LDHs catalyst.

**Figure 6 molecules-26-00395-f006:**
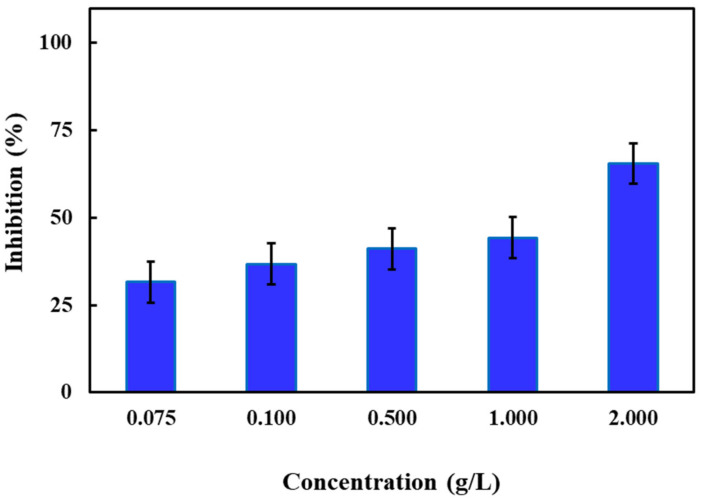
Mean growth inhibition (± SD; *n* = 7) of *S. polyrhiza* after 72 h of exposure to concentrations of 0.075, 0.10, 0.50, 1.00, and 2.00 g/L Zn-Fe LDHs catalyst.

## Data Availability

Data are contained within the article or Appendix A.

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
