# Peer review of "Toxicity of Zn-Fe Layered Double Hydroxide to Different Organisms in the Aquatic Environment"

_molecules, 2021, doi:10.3390/molecules26020395_

Round 1
Reviewer 1 Report
The Authors presented the preparation of a Zn-Fe LDH-material, aiming at the assessment of its toxicity. The work presents a quite solid introduction, although very focused over Zn-Fe LDHs instead of offering in principle a wider vision on the topic and a justification of their specific interest in Zn-Fe materials (cheap precursors, simple synthetic procedures...?). The Materials & Methods section is full of information and looks to be well organized. Concerning the discussion of the results, the characterization is well discussed, but there was a higher expectation for the sections dealing with the toxicity assays. The Authors honestly claim that their approach offers, compared to the literature, a wider vision over LDHs toxicological aspects since different kinds of assays are performed for the material, considering four different subject organisms. Nevertheless, the discussion results in a sequence of numbers derived from the experiments and, when it should be the time to offer a solid answer to the reasons why their material acts in a certain way, there are just speculations. Citing the last sentence from section 3.2.2 (but it is not just here), ".. to gain further insight into the toxicity pathway and potential impacts of Zn-Fe LDH catalyst on microalgae, characterization of the catalyst material before and after exposure to the test medium could be important". And this is the weak point since it was needed to recollect the material and plan a couple of experiments, in order to provide an answer and a novelty. In my opinion, the Authors should plan a couple of experiments for their materials, offer those conclusions, so that the picture is complete. Eventually, if they consider that it is not possible to run those experiments on the materials after the toxicological tests, they should report the reasons why: Only in this case the work could be really considered "over".
Then, there are here and there corrections/suggestions, which I hope the Authors could use in order to smooth some aspects and/or correct some occasional errors.
***
Concerning the Supporting Information, minor correction should be done (correct labelling, table layout, ...).
***
Concerning the word file collecting non published results, it looks to be a double version of the SI file. Please remove it and, as mentioned above, remove any connection in the main text to those data, or better mention the proper figure/table in the SI file.
All my notes/suggestions are reported in the .pdf file in attachment; single passages are evidenced in yellow, and the comment therein reported.
Overall, I consider that the research work is valid and could appear more solid if some suggestion would be followed, as well as an improved discussion of the results and the collection of some extra experimental data.

Author Response
Ms. Ref. No.: Molecules-1037547
Title: Toxicity of Zn-Fe layered double hydroxide to different organisms in the aquatic environment
Journal: Molecules
Authors: Olga Koba Ucun, TuÄŸba Ölmez Hanci, Idil Arslan-Alaton, Samira Arefi-Oskoui, Alireza Khataeeb, Mehment Kobya, Yasin Orooji.
Dear Editor,
First of all, thank you very much for considering our manuscript for possible publication in the Journal of “Molecules”. After carefully considering the reviewers comments, the following responses have been prepared according to the comments of the reviewers. We also appreciate the reviewers for their useful comments. We have indicated the revised parts in the text with blue font.
Sincerely yours,
Alireza Khataee (PhD)
Corresponding author
Reviewer 1,
The Authors presented the preparation of a Zn-Fe LDH-material, aiming at the assessment of its toxicity. The work presents a quite solid introduction, although very focused over Zn-Fe LDHs instead of offering in principle a wider vision on the topic and a justification of their specific interest in Zn-Fe materials (cheap precursors, simple synthetic procedures...?). The Materials & Methods section is full of information and looks to be well organized. Concerning the discussion of the results, the characterization is well discussed, but there was a higher expectation for the sections dealing with the toxicity assays. The Authors honestly claim that their approach offers, compared to the literature, a wider vision over LDHs toxicological aspects since different kinds of assays are performed for the material, considering four different subject organisms. Nevertheless, the discussion results in a sequence of numbers derived from the experiments and, when it should be the time to offer a solid answer to the reasons why their material acts in a certain way, there are just speculations. Citing the last sentence from section 3.2.2 (but it is not just here), ".. to gain further insight into the toxicity pathway and potential impacts of Zn-Fe LDH catalyst on microalgae, characterization of the catalyst material before and after exposure to the test medium could be important". And this is the weak point since it was needed to recollect the material and plan a couple of experiments, in order to provide an answer and a novelty. In my opinion, the Authors should plan a couple of experiments for their materials, offer those conclusions, so that the picture is complete. Eventually, if they consider that it is not possible to run those experiments on the materials after the toxicological tests, they should report the reasons why: Only in this case the work could be really considered "over".
Response: we respect the comment of the reviewer. In the present paper, it was aimed at evaluating the acute toxicity of a novel catalyst material using commercially available toxicity test kits. The test organisms and test kits have been selected considering their ease of use, relatively low cost, but in particular, because they are very sensitive, well–known, widely used, standard/reference test methods so that the authors could safely use them in their tests. The major purpose of the toxicity section was not to enlighten the toxicity pathway (which is known for the materials as well as the organisms used in these tests), but to show the applicants how commercial toxicity tests can be employed, how they respond to the dose of the catalyst materials, as well as the practical problems being faced during all stages of the test; namely “heterogeneous” sample preparation, incubation, recording and evaluating toxicity results (dose-response data). Slight modifications were proposed that enabled the proper evaluation of the best results.
So far, the authors have experienced several toxicity tests protocols, test methods and acute/chronic toxicity test procedures in the past. Their present study highlighted the fact that some test organisms are more sensitive and hence appropriate for testing solid catalyst materials. Further, again, it could be demonstrated that a battery test should be preferred to a single test since the suitability of the organism for testing solid nanocatalyst differs from routine applications and hence special precautions should be taken prior to and during acute testing.
In addition, the following references which were missed in the reference lists were added:
- Aruoja, V., Dubourguier, H.-C., Kasemets, K., Kahru, A., 2009. Toxicity of nanoparticles of CuO, ZnO and TiO2 to microalgae Pseudokirchneriella subcapitata. Sci. Total Environ. 407, 1461–1468. https://doi:10.1016/j.scitotenv.2008.10.053.
- Bacchetta, R., Maran, B., Marelli, M., Santo, N., Tremolada, P., 2016. Role of soluble zinc in ZnO nanoparticle cytotoxicity in Daphnia magna: a morphological approach. Environ. Res. 148, 376–385. https://doi.org/10.1016/j.envres.2016.04.028.
- Blinova, I., Ivask, A., Heinlaan, M., Mortimer, M., Kahru, A., 2010. Ecotoxicity of nanoparticles of CuO and ZnO in natural water. Environ. Pollut. 158, 41–47. https://doi.org/10.1016/j.envpol.2009.08.017.
- Franklin, N.M., Rogers, N.J., Apte, S.C., Batley, G.E., Gadd, G.E., Casey, P.S., 2007. Comparative Toxicity of Nanoparticulate ZnO, Bulk ZnO, and ZnCl2 to a Freshwater Microalga (Pseudokirchneriella subcapitata): The Importance of Particle Solubility. Environ. Sci. 41, 8484–8490. https://doi.org/10.1021/es071445r.
- Wang, Z., Yu, H., Ma, K., Chen, Y., Zhang, X., Wang, T., Li, S., Zhu, X., Wang, X., 2018. Flower-like surface of three-metal-component layered double hydroxide composites for improved antibacterial activity of lysozyme. Bioconjugate Chem. 29, 2090-2099. https://doi.org/10.1021/acs.bioconjchem.8b00305.
- Then, there are here and there corrections/suggestions, which I hope the Authors could use in order to smooth some aspects and/or correct some occasional errors.
Response: We thank the reviewer for the valuable comments. The comments and the suggestions of the reviewer are considered to improve the manuscript.
***
Concerning the Supporting Information, minor correction should be done (correct labelling, table layout, ...).
Response: We respect the comment of the reviewer. The label of the table and the figure was corrected in supporting information.
***
- Concerning the word file collecting non published results, it looks to be a double version of the SI file. Please remove it and, as mentioned above, remove any connection in the main text to those data, or better mention the proper figure/table in the SI file.
Response: We respect the comment of the reviewer. The comment was considered.
- All my notes/suggestions are reported in the .pdf file in attachment; single passages are evidenced in yellow, and the comment therein reported.
The comments of reviewer 1 in PDF file are:
- I do not find appropriate to put scientific articles instead of reviews when the Authors firstly introduce the topic of their researc. Or better, 3out of 4 articles are not reviews, which is not satisfying when you speak aboutsuch well-established investigation fiels. The Authors are kindly asked to put more references here.
Please introduce the following references here and in the ext lines of the introduction (DOIs are listed):
10.1016/j.clay.2017.12.021, 10.3390/cryst9070361, 10.3390/nano8100747, 10.1039/C8TA11273H, 10.1039/C6TA01668E, 10.1039/C5CC07296D
Response: We respect the comment of the reviewer. We thank the reviewer for introducing valuable references. The aforementioned references were used to improve the manuscript.
- In my eyes, every single application field should have at least one reference if not 2 and, more importantly, the authors cannot just put refs. 6-8 (which are related to their Zn-Fe materials) but, since it is a sentence which should describe LDHs whole applications scenario, they have to expand their list of citations. Here articles and reviews can be used as well.
Response: We respect to the comment of the reviewer. Some appropriate references were added.
- In this case, it would be more appropriate to write as "LDHs". This is not the only case in the text, please go through it and fix the wording.
Response: We thank the reviewer for the comment. We replaced “LDH” with “LDHs” in overall the manuscript.
- Although it is quite common, I would rather prefer to see a different symbol to be used, and report iron chloride as FeCl3· 6H2O instead of FeCl3.6H2O.
Response: We thank the reviewer for this comment. We replaced “FeCl3.6H2O” with “FeCl3· 6H2O”.
- Considering that the Authors are reporting a single significant figure, I would suggest to report the number of millimoles, and at least have 3 significant figures (x.xx millimoles).
Response: We respect the comment of the reviewer. “0.001 mol and 0.03 mol” were replaced with “1.00 mmol and 3.00 mmol”.
- There is space missing in “topH”.
Response: We thank the reviewer for the comment. “topH” was replaced with “to pH”.
- Wrong splelling, please correct it (the Netherlands)
Response: We thank the reviewer for the comment. We revised it.
- Please introduced the well-established short name FT-IR, as you do in the Abstract.
Response: We thank the reviewer for the comment. We added “FT-IR” after “Fourier transform infrared spectrometer” as short name.
- The citation in “purchased from MicroBioTests Inc. company [34]” is not proper (a ISO standard paper is reported while here the Authors are just enumerating the products they've used). Please remove it, or anyway please doublecheck it.
Response: We thank the reviewer for the comment. The reference was omitted.
- I think the Authors wanted to write "... and all the samples pH values were...". Please double check and correct the phrasing
Response: We thank the reviewer for this comment. This sentence was revised.
- I found the temperature values reported as x°C and x °C. In my opinion, the second form should be the correct one. Please, uniform the style within the whole text.
Response: We thank the reviewer for comment. We used “x °C” style in overall the manuscript.
- Please correct “pipet”.
Response: We thank the reviewer for comment. “pipet” was replaced with “pipette”,
- In sentence of “The test areas were measured with the Image J software”, citation missing. Please refer to https://imagej.net/Citing.
Response: We thank the reviewer for this comment. The proposed reference was added.
- In sentence of “to the carbonate (CO32-) groups” I would rather say anions, since they're not a functional group. Please correct it.
Response: We thank the reviewer for the comment. We replaced “groups” with “anions”.
- Considering the broadness of the band, don't you believe that an enormous contribution is given by the stabilizing interlayer water molecules, as well? Please cross-check, and eventually modify the sentence accordingly with what said before
Response: We thank the reviewer for comment. The sentence was corrected in manuscript.
- Have you done any statistics before reporting the value of average thickness of the synthesized LDHs? Which is its standard deviation? Please report it.
Response: We respect the comment of the reviewer. The thickness of the synthesized Zn-Fe LDHs was determined using SEM images and Digimizer software, and the average thickness was reported. The average thickness of the synthesized LDHs was determined 45 ± 0.6 nm.
- The font of “V. fischeri” should be changed as italic, in order to be homogeneous within the whole text.
Response: We thank the reviewer for the comment. The font of “V. fischeri” was changed to italic in the manuscript.
- Here, the Authors should immediately say whether they are speaking or not of the 3h-prepared sample, instead of waiting for the next 5 livens. Moreover, it would be the section more readable if a table is inserted, and fulled with all the inhibition percentafges andEC50 values.
Response: As reviewer suggested, a table comparing the resultant EC50 values is reported. The toxicity test results obtained with the selected bioassays (expressed as EC50 values) are summarized in Table 1. From Table 1 it is apparent that the tested catalyst is highly toxic (inhibitory) towards the test organisms in its typical working range.
Table 1. Summary of test results obtained for the Zn-Fe LDH catalyst being subjected to different test organisms.
|
Test Organism |
Sample preparation time |
Test exposure time |
EC50 (g/L) |
|
V. fischeri |
3 h |
15 min |
1.20±0.12 |
|
30 min |
0.32±0.03 |
||
|
V. fischeri |
1 h |
15 min |
0.31±0.03 |
|
30 min |
0.11±0.01 |
||
|
P. subcapitata |
Prior to test |
72 h |
a |
|
D. magna |
Prior to test |
24 h |
0.17±0.02 |
|
48 h |
0.10±0.01 |
||
|
S. polyrhiza |
Prior to test |
72 h |
0.81±0.19 |
a All tested concentrations were toxic towards the test organism.
- In my opinion, the Authors should take a more straight position, mentioning the reason why one material is more toxic than the other, and eventually supporting their statement with some references
Response: The toxicity of a substance can be affected by many different factors, such as the pathway of administration, exposure (incubation) time, the physical form/state and structure of the toxicant as well as the type/nature of the test organism.
In the present study, the catalyst structure was not investigated after the toxicity tests. Eventually, metal leaching and slight surface modifications could be expected during incubation. Besides, a change in acute toxicity values with varying incubation time is always possible and cannot be solely attributed to the physicochemical properties of the catalyst material or test organism. Typically, incubation time increases the acute toxicity since the exposure time is increased with incubation time.
In order to prevent catalyst agglomeration and precipitation of the test medium’s salts onto the catalyst, the sample preparation time was reduced/minimized to 3 hours as mentioned in the manuscript. It should be noted that preparation and incubation time are different variables that should be considered separately. Catalyst preparation time was modified because of the specific situation of working with a solid catalyst material.
[1] https://web.archive.org/web/20181001211259/http://alttox.org/mapp/toxicity-endpoints-tests/
[2] Bhattacharya S, Zhang Q, Carmichael PL, Boekelheide K, Andersen ME. Toxicity testing in the 21 century: defining new risk assessment approaches based on perturbation of intracellular toxicity pathways. PLoS ONE (2011) 6:e20887. doi: 10.1371/journal.pone.0020887
[3] Dong Z, Liu Y, Duan L, Bekele D, Naidu R. Uncertainties in human health risk assessment of environmental contaminants: a review and perspective. Environ Int. (2015) 85:120–32. doi: 10.1016/j.envint.2015.09.008
- “Fe3+” is not necessary in “mg Fe3+/L”, here it is obvious that you are talking about Fe3+. To be removed.
Response: We thank the reviewer for the comment. The unit was revised in manuscript.
- There is a mismatch in the number of significant figures in the measured value and the deviation... please report homogeneous and consistent numbers (e.g., 0.17±0.02).
Response: We thank the reviewer for the valuable suggestion. The numbers and deviation were rechecked and revised in the manuscript.
- Please correct the labels. This should be Table S1 in the SI file.
Response: We respect the comment of the reviewer. The labels were revised in the manuscript and - supplementary information file.
- The section of “Supplementary Materials” was not correctly updated. Please fix this issue.
Response: We respect to the comment of reviewer. The revised file of Supplementary Materials is available.
- The section of “Author contributions” was not correctly updated. Please fix this issue.
Response: We respect t the comment of reviewer. The information about the author contributions was added to the file of Credit. Authors’ credit statements:
Olga Koba Ucun: Reviewing and editing, experiments, characterization of samples; TuÄŸba Ölmez-Hanci: Reviewing and Editing; Idil Arslan-Alaton: Supervision, reviewing and editing; Samira Arefi-Oskoui: Experiments, characterization of samples, reviewing and editing; Alireza Khataee: Supervision, reviewing and editing; Mehment Kobya: Reviewing and editing; Yasin Orooji: Resources, characterization of samples.
- The section of “Funding” was not correctly updated. Please fix this issue.
Response: We thank the reviewer for the comment. The authors are thankful for the financial support of Istanbul Technical University under project Nr. MAB-2019-42237.
- The section of “Conflicts of Interest” was not correctly updated. Please fix this issue.
Response: Olga Koba Ucun: Reviewing and editing, experiments, characterization of samples; TuÄŸba Ölmez-Hanci: Reviewing and Editing; Idil Arslan-Alaton: Supervision, reviewing and editing; Samira Arefi-Oskoui: Experiments, characterization of samples, reviewing and editing; Alireza Khataee: Supervision, reviewing and editing; Mehment Kobya: Reviewing and editing; Yasin Orooji: Resources, characterization of samples.
- Missing the place of the publication (I presume Geneva, like te others), and the year in the reference of “International Organization for Standardization (ISO) 6341, 2012. Water quality – Determination of the inhibition of the mobility of Daphnia magna Straus (Cladocera, Crustacea)– Acute toxicity test”.
Response: We thank the reviewer for the comment. The reference was revised.
- I presume the reference “Bulich, A.A. A practical and reliable method for monitoring the toxicity of aquatic samples. 1982.” is a book. Pleas check the citation (missing the editor and the place where it is based, etc.)
Response: We thank the reviewer for the comment. The aforementioned reference was rechecked. It is a research article, and the volume number and the page number were added.
- The name in reference of “de OF Rossetto, A.L.; Melegari, S.P.; Ouriques, L.C.; Matias, W.G. Comparative evaluation of acute and chronic toxicities of CuO nanoparticles and bulk using Daphnia magna and Vibrio fischeri. Sci. Total Environ. 2014, 490, 807-814. “is wrongly cited. "OF" should be "O. F.". Please double check this issue.
Response: We thank the reviewer for the comment. The name of “de OF Rossetto, A.L” was changed to “Rossetto, A.L.D.F.”.
Overall, I consider that the research work is valid and could appear more solid if some suggestion would be followed, as well as an improved discussion of the results and the collection of some extra experimental data.

Reviewer 2 Report
The authors mention "study of Zhao et al." (p.2, line 56). Why an exact reference is not provided in the list of references?
Peaks indexing in XRD pattern should be checked thoroughly. The first peak is usually attributed to (003) plain (not 001), the following peaks also have doubtful attribution.
XRD pattern gives a feeling of low crystallinity of the LDH sample and a presence of extraneous phase in it. SEM images don't contradict the supposition since there are nongeometrical amorphous fragments in them. It requires some comments. Could the extraneous phase affect toxicity?
In the experiments with V. fischery reason of EC50 decrease with an increase of contact time is unclear.
There are minor misprints in the manuscript (e.g., line 269)
Author Response
Ms. Ref. No.: Molecules-1037547
Title: Toxicity of Zn-Fe layered double hydroxide to different organisms in the aquatic environment
Journal: Molecules
Authors: Olga Koba Ucun, TuÄŸba Ölmez Hanci, Idil Arslan-Alaton, Samira Arefi-Oskoui, Alireza Khataeeb, Mehment Kobya, Yasin Orooji.
Dear Editor,
First of all, thank you very much for considering our manuscript for possible publication in the Journal of “Molecules”. After carefully considering the reviewers comments, the following responses have been prepared according to the comments of the reviewers. We also appreciate the reviewers for their useful comments. We have indicated the revised parts in the text with blue font.
Sincerely yours,
Alireza Khataee (PhD)
Corresponding author
Reviewer 2
The authors mention "study of Zhao et al." (p.2, line 56). Why an exact reference is not provided in the list of references?
- The authors mention "study of Zhao et al." (p.2, line 56). Why an exact reference is not provided in the list of references?
Response: We thank the reviewer for the valuable comment. The reference was added to the list of references.
- Peaks indexing in XRD pattern should be checked thoroughly. The first peak is usually attributed to (003) plain (not 001), the following peaks also have doubtful attribution.
Response: We thank the reviewer for the valuable comment. The peaks indexing in XRD pattern were revised. The peaks centered at 2θ of 12.1°, 23.3°, 34.5°, 39.0°, 45.9°, 59.7°, and 61.2° can be attributed to the 003, 006, 009, 015, 018, 110, 113 and 113 crystalline plane of the synthesized sample, indicating the successful synthesis of LDHs
- XRD pattern gives a feeling of low crystallinity of the LDH sample and a presence of extraneous phase in it. SEM images don't contradict the supposition since there are nongeometrical amorphous fragments in them. It requires some comments. Could the extraneous phase affect toxicity?
Response: We respect the comment of the reviewer. The specific strong and sharp diffraction peak at 2θ value of 12.1° (003) is assigned to the LDH structure, confirming the high crystallinity of the LDH. Moreover, other index diffraction peaks centered at 2θ of 23.3°, 34.5°, 39.0°, 45.9°, 59.7°, and 61.2° confirm the successful synthesis of LDHs [1-3]. In addition, the SEM images and FT-IR spectrum corroborate the successful formation of Zn-Fe LDHs via the co-precipitation method. These results of the analysis show that the formed phase can be mainly attributed to the layered double hydroxide, and the presence of extraneous phase is negligible which can not affect the toxicity tests.
[1] Rahmanian, O., S. Amini, and M. Dinari, Preparation of zinc/iron layered double hydroxide intercalated by citrate anion for capturing Lead (II) from aqueous solution. Journal of Molecular Liquids, 2018. 256: p. 9-15.
[2] Parida, K. and L. Mohapatra, Carbonate intercalated Zn/Fe layered double hydroxide: a novel photocatalyst for the enhanced photo degradation of azo dyes. Chemical engineering journal, 2012.
179: p. 131-139.
[3] Zaher, A., et al., Zn/Fe LDH as a clay-like adsorbent for the removal of oxytetracycline from water: combining experimental results and molecular simulations to understand the removal mechanism. Environmental Science and Pollution Research, 2020: p. 1-14.
- In the experiments with V. fischery reason of EC50 decrease with an increase of contact time is unclear.
Response: We respect the comment of the reviewer. The toxicity / relative inhibition increased with incubation time since the exposure time negatively affects the tested organisms. This was also evidenced in the present study.
- There are minor misprints in the manuscript (e.g., line 269)
Response: We respect the comment of the reviewer. The manuscript was rechecked and the misprints were revised in overall the manuscript.

Round 2
Reviewer 1 Report
I gratefully thank the Authors for the points of discussion they've offered in their reply. I am glad to see that they have positively answered most of my questions/issues. The manuscript is more solid and enjoyable for the readership; for this reason, I'm in favor of the publication of this newer version.
I do not have any further issues to discuss or corrections to point out.
Good luck with your future research activities, I wish you a happy New Year.